# Linear Response Methods for Accurate Covariance Estimates from Mean Field Variational Bayes

**Ryan Giordano**
UC Berkeley
rgiordano@berkeley.edu

Tamara Broderick
MIT
tbroderick@csail.mit.edu

Michael Jordan
UC Berkeley
jordan@cs.berkeley.edu

## Abstract

Mean field variational Bayes (MFVB) is a popular posterior approximation method due to its fast runtime on large-scale data sets. However, a well known major failing of MFVB is that it underestimates the uncertainty of model variables (sometimes severely) and provides no information about model variable covariance. We generalize linear response methods from statistical physics to deliver accurate uncertainty estimates for model variables—both for individual variables and coherently across variables. We call our method *linear response variational Bayes* (LRVB). When the MFVB posterior approximation is in the exponential family, LRVB has a simple, analytic form, even for non-conjugate models. Indeed, we make no assumptions about the form of the true posterior. We demonstrate the accuracy and scalability of our method on a range of models for both simulated and real data.

## 1    Introduction

With increasingly efficient data collection methods, scientists are interested in quickly analyzing ever larger data sets. In particular, the promise of these large data sets is not simply to fit old models but instead to learn more nuanced patterns from data than has been possible in the past. In theory, the Bayesian paradigm yields exactly these desiderata. Hierarchical modeling allows practitioners to capture complex relationships between variables of interest. Moreover, Bayesian analysis allows practitioners to quantify the uncertainty in any model estimates—and to do so coherently across all of the model variables.

*Mean field variational Bayes* (MFVB), a method for approximating a Bayesian posterior distribution, has grown in popularity due to its fast runtime on large-scale data sets [1–3]. But a well known major failing of MFVB is that it gives underestimates of the uncertainty of model variables that can be arbitrarily bad, even when approximating a simple multivariate Gaussian distribution [4–6]. Also, MFVB provides no information about how the uncertainties in different model variables interact [5–8].

By generalizing linear response methods from statistical physics [9–12] to exponential family variational posteriors, we develop a methodology that augments MFVB to deliver accurate uncertainty estimates for model variables—both for individual variables and coherently across variables. In particular, as we elaborate in Section 2, when the approximating posterior in MFVB is in the exponential family, MFVB defines a fixed-point equation in the means of the approximating posterior,

and our approach yields a covariance estimate by perturbing this fixed point. We call our method *linear response variational Bayes* (LRVB).

We provide a simple, intuitive formula for calculating the linear response correction by solving a linear system based on the MFVB solution (Section 2.2). We show how the sparsity of this system for many common statistical models may be exploited for scalable computation (Section 2.3). We demonstrate the wide applicability of LRVB by working through a diverse set of models to show that the LRVB covariance estimates are nearly identical to those produced by a Markov Chain Monte Carlo (MCMC) sampler, even when MFVB variance is dramatically underestimated (Section 3). Finally, we focus in more depth on models for finite mixtures of multivariate Gaussians (Section 3.3), which have historically been a sticking point for MFVB covariance estimates [5, 6]. We show that LRVB can give accurate covariance estimates orders of magnitude faster than MCMC (Section 3.3). We demonstrate both theoretically and empirically that, for this Gaussian mixture model, LRVB scales linearly in the number of data points and approximately cubically in the dimension of the parameter space (Section 3.4).

**Previous Work.**   Linear response methods originated in the statistical physics literature [10–13]. These methods have been applied to find new learning algorithms for Boltzmann machines [13], covariance estimates for discrete factor graphs [14], and independent component analysis [15]. [16] states that linear response methods could be applied to general exponential family models but works out details only for Boltzmann machines. [10], which is closest in spirit to the present work, derives general linear response corrections to variational approximations; indeed, the authors go further to formulate linear response as the first term in a functional Taylor expansion to calculate full pairwise joint marginals. However, it may not be obvious to the practitioner how to apply the general formulas of [10]. Our contributions in the present work are (1) the provision of concrete, straightforward formulas for covariance correction that are fast and easy to compute, (2) demonstrations of the success of our method on a wide range of new models, and (3) an accompanying suite of code.

## 2   Linear response covariance estimation

### 2.1   Variational Inference

Suppose we observe $N$ data points, denoted by the $N$-long column vector $x$, and denote our unobserved model parameters by $\theta$. Here, $\theta$ is a column vector residing in some space $\Theta$; it has $J$ subgroups and total dimension $D$. Our model is specified by a distribution of the observed data given the model parameters—the likelihood $p(x|\theta)$—and a prior distributional belief on the model parameters $p(\theta)$. Bayes' Theorem yields the posterior $p(\theta|x)$.

Mean-field variational Bayes (MFVB) approximates $p(\theta|x)$ by a factorized distribution of the form $q(\theta) = \prod_{j=1}^{J} q(\theta_j)$. $q$ is chosen so that the Kullback-Liebler divergence $\mathrm{KL}(q||p)$ between $q$ and $p$ is minimized. Equivalently, $q$ is chosen so that $E := L + S$, for $L := \mathbb{E}_q[\log p(\theta|x)]$ (the expected log posterior) and $S := -\mathbb{E}_q[\log q(\theta)]$ (the entropy of the variational distribution), is maximized:

$$q^* := \arg\min_q \mathrm{KL}(q||p) = \arg\min_q \mathbb{E}_q\left[\log q(\theta) - \log p(\theta|x)\right] = \arg\max_q E. \qquad (1)$$

Up to a constant in $\theta$, the objective $E$ is sometimes called the "evidence lower bound", or the ELBO [5]. In what follows, we further assume that our variational distribution, $q(\theta)$, is in the exponential family with natural parameter $\eta$ and log partition function $A$: $\log q(\theta|\eta) = \eta^T \theta - A(\eta)$ (expressed with respect to some base measure in $\theta$). We assume that $p(\theta|x)$ is expressed with respect to the same base measure in $\theta$ as for $q$. Below, we will make only mild regularity assumptions about the true posterior $p(\theta|x)$ and no assumptions about its form.

If we assume additionally that the parameters $\eta^*$ at the optimum $q^*(\theta) = q(\theta|\eta^*)$ are in the interior of the feasible space, then $q(\theta|\eta)$ may instead be described by the mean parameterization: $m := \mathbb{E}_q \theta$

with $m^* := \mathbb{E}_{q^*}\theta$. Thus, the objective $E$ can be expressed as a function of $m$, and the first-order condition for the optimality of $q^*$ becomes the fixed point equation

$$\left.\frac{\partial E}{\partial m}\right|_{m=m^*} = 0 \;\Leftrightarrow\; \left.\left(\frac{\partial E}{\partial m}+m\right)\right|_{m=m^*} = m^* \;\Leftrightarrow\; M(m^*) = m^* \text{ for } M(m) := \frac{\partial E}{\partial m}+m. \quad (2)$$

## 2.2 Linear Response

Let $V$ denote the covariance matrix of $\theta$ under the variational distribution $q^*(\theta)$, and let $\Sigma$ denote the covariance matrix of $\theta$ under the true posterior, $p(\theta|x)$:

$$V := \text{Cov}_{q^*}\theta, \qquad \Sigma := \text{Cov}_p\theta.$$

In MFVB, $V$ may be a poor estimator of $\Sigma$, even when $m^* \approx \mathbb{E}_p\theta$, i.e., when the marginal estimated means match well [5–7]. Our goal is to use the MFVB solution and linear response methods to construct an improved estimator for $\Sigma$. We will focus on the covariance of the natural sufficient statistic $\theta$, though the covariance of functions of $\theta$ can be estimated similarly (see Appendix A).

The essential idea of linear response is to perturb the first-order condition $M(m^*) = m^*$ around its optimum. In particular, define the distribution $p_t(\theta|x)$ as a log-linear perturbation of the posterior:

$$\log p_t(\theta|x) \quad := \quad \log p(\theta|x) + t^T\theta - C(t), \quad (3)$$

where $C(t)$ is a constant in $\theta$. We assume that $p_t(\theta|x)$ is a well-defined distribution for any $t$ in an open ball around 0. Since $C(t)$ normalizes $p_t(\theta|x)$, it is in fact the cumulant-generating function of $p(\theta|x)$, so the derivatives of $C(t)$ evaluated at $t=0$ give the cumulants of $\theta$. To see why this perturbation may be useful, recall that the second cumulant of a distribution is the covariance matrix, our desired estimand:

$$\Sigma = \text{Cov}_p(\theta) = \left.\frac{d}{dt^T dt}C(t)\right|_{t=0} = \left.\frac{d}{dt^T}\mathbb{E}_{p_t}\theta\right|_{t=0}.$$

The practical success of MFVB relies on the fact that its estimates of the mean are often good in practice. So we assume that $m_t^* \approx \mathbb{E}_{p_t}\theta$, where $m_t^*$ is the mean parameter characterizing $q_t^*$ and $q_t^*$ is the MFVB approximation to $p_t$. (We examine this assumption further in Section 3.) Taking derivatives with respect to $t$ on both sides of this mean approximation and setting $t=0$ yields

$$\Sigma = \text{Cov}_p(\theta) \approx \left.\frac{dm_t^*}{dt^T}\right|_{t=0} =: \hat{\Sigma}, \quad (4)$$

where we call $\hat{\Sigma}$ the *linear response variational Bayes* (LRVB) estimate of the posterior covariance of $\theta$.

We next show that there exists a simple formula for $\hat{\Sigma}$. Recalling the form of the KL divergence (see Eq. (1)), we have that $-\text{KL}(q\|p_t) = E + t^T m =: E_t$. Then by Eq. (2), we have $m_t^* = M_t(m_t^*)$ for $M_t(m) := M(m) + t$. It follows from the chain rule that

$$\frac{dm_t^*}{dt} = \left.\frac{\partial M_t}{\partial m^T}\right|_{m=m_t^*}\frac{dm_t^*}{dt} + \frac{\partial M_t}{\partial t} = \left.\frac{\partial M_t}{\partial m^T}\right|_{m=m_t^*}\frac{dm_t^*}{dt} + I, \quad (5)$$

where $I$ is the identity matrix. If we assume that we are at a strict local optimum and so can invert the Hessian of $E$, then evaluating at $t=0$ yields

$$\hat{\Sigma} = \left.\frac{dm_t^*}{dt^T}\right|_{t=0} = \frac{\partial M}{\partial m}\hat{\Sigma} + I = \left(\frac{\partial^2 E}{\partial m \partial m^T} + I\right)\hat{\Sigma} + I \quad\Rightarrow\quad \hat{\Sigma} = -\left(\frac{\partial^2 E}{\partial m \partial m^T}\right)^{-1}, \quad (6)$$

where we have used the form for $M$ in Eq. (2). So the LRVB estimator $\hat{\Sigma}$ is the negative inverse Hessian of the optimization objective, $E$, as a function of the mean parameters. It follows from Eq. (6) that $\hat{\Sigma}$ is both symmetric and positive definite when the variational distribution $q^*$ is at least a local maximum of $E$.

We can further simplify Eq. (6) by using the exponential family form of the variational approximating distribution $q$. For $q$ in exponential family form as above, the negative entropy $-S$ is dual to the log partition function $A$ [17], so $S = -\eta^T m + A(\eta)$; hence,

$$\frac{dS}{dm} = \frac{\partial S}{\partial \eta^T} \frac{d\eta}{dm} + \frac{\partial S}{\partial m} = \left( \frac{\partial A}{\partial \eta} - m \right) \frac{d\eta}{dm} - \eta(m) = -\eta(m).$$

Recall that for exponential families, $\partial \eta(m)/\partial m = V^{-1}$. So Eq. (6) becomes[1]

$$\hat{\Sigma} = - \left( \frac{\partial^2 L}{\partial m \partial m^T} + \frac{\partial^2 S}{\partial m \partial m^T} \right)^{-1} = -(H - V^{-1})^{-1}, \text{ for } H := \frac{\partial^2 L}{\partial m \partial m^T}. \Rightarrow$$

$$\hat{\Sigma} = (I - VH)^{-1}V. \tag{7}$$

When the true posterior $p(\theta|x)$ is in the exponential family and contains no products of the variational moment parameters, then $H = 0$ and $\hat{\Sigma} = V$. In this case, the mean field assumption is correct, and the LRVB and MFVB covariances coincide at the true posterior covariance. Furthermore, even when the variational assumptions fail, as long as certain mean parameters are estimated exactly, then this formula is also exact for covariances. E.g., notably, MFVB is well-known to provide arbitrarily bad estimates of the covariance of a multivariate normal posterior [4–7], but since MFVB estimates the means exactly, LRVB estimates the covariance exactly (see Appendix B).

### 2.3 Scaling the matrix inverse

Eq. (7) requires the inverse of a matrix as large as the parameter dimension of the posterior $p(\theta|x)$, which may be computationally prohibitive. Suppose we are interested in the covariance of parameter sub-vector $\alpha$, and let $z$ denote the remaining parameters: $\theta = (\alpha, z)^T$. We can partition $\Sigma = (\Sigma_\alpha, \Sigma_{\alpha z}; \Sigma_{z\alpha}, \Sigma_z)$. Similar partitions exist for $V$ and $H$. If we assume a mean-field factorization $q(\alpha, z) = q(\alpha)q(z)$, then $V_{\alpha z} = 0$. (The variational distributions may factor further as well.) We calculate the Schur complement of $\hat{\Sigma}$ in Eq. (7) with respect to its $z$th component to find that

$$\hat{\Sigma}_\alpha = \left( I_\alpha - V_\alpha H_\alpha - V_\alpha H_{\alpha z} \left( I_z - V_z H_z \right)^{-1} V_z H_{z\alpha} \right)^{-1} V_\alpha. \tag{8}$$

Here, $I_\alpha$ and $I_z$ refer to $\alpha$- and $z$-sized identity matrices, respectively. In cases where $(I_z - V_z H_z)^{-1}$ can be efficiently calculated (e.g., all the experiments in Section 3; see Fig. (5) in Appendix D), Eq. (8) requires only an $\alpha$-sized inverse.

## 3 Experiments

We compare the covariance estimates from LRVB and MFVB in a range of models, including models both with and without conjugacy [2]. We demonstrate the superiority of the LRVB estimate to MFVB in all models before focusing in on Gaussian mixture models for a more detailed scalability analysis.

For each model, we simulate datasets with a range of parameters. In the graphs, each point represents the outcome from a single simulation. The horizontal axis is always the result from an MCMC

procedure, which we take as the ground truth. As discussed in Section 2.2, the accuracy of the LRVB covariance for a sufficient statistic depends on the approximation $m_t^* \approx \mathbb{E}_{p_t}\theta$. In the models to follow, we focus on regimes of moderate dependence where this is a reasonable assumption for most of the parameters (see Section 3.2 for an exception). Except where explicitly mentioned, the MFVB means of the parameters of interest coincided well with the MCMC means, so our key assumption in the LRVB derivations of Section 2 appears to hold.

## 3.1 Normal-Poisson model

**Model.** First consider a Poisson generalized linear mixed model, exhibiting non-conjugacy. We observe Poisson draws $y_n$ and a design vector $x_n$, for $n = 1, ..., N$. Implicitly below, we will everywhere condition on the $x_n$, which we consider to be a fixed design matrix. The generative model is:

$$z_n|\beta, \tau \overset{indep}{\sim} \mathcal{N}\left(z_n|\beta x_n, \tau^{-1}\right), \quad y_n|z_n \overset{indep}{\sim} \text{Poisson}\left(y_n|\exp(z_n)\right), \quad (9)$$

$$\beta \sim \mathcal{N}(\beta|0, \sigma_\beta^2), \quad \tau \sim \Gamma(\tau|\alpha_\tau, \beta_\tau).$$

For MFVB, we factorize $q(\beta, \tau, z) = q(\beta)\, q(\tau) \prod_{n=1}^N q(z_n)$. Inspection reveals that the optimal $q(\beta)$ will be Gaussian, and the optimal $q(\tau)$ will be gamma (see Appendix D). Since the optimal $q(z_n)$ does not take a standard exponential family form, we restrict further to Gaussian $q(z_n)$. There are product terms in $L$ (for example, the term $\mathbb{E}_q[\tau]\,\mathbb{E}_q[\beta]\,\mathbb{E}_q[z_n]$), so $H \neq 0$, and the mean field approximation does not hold; we expect LRVB to improve on the MFVB covariance estimate. A detailed description of how to calculate the LRVB estimate can be found in Appendix D.

**Results.** We simulated 100 datasets, each with 500 data points and a randomly chosen value for $\mu$ and $\tau$. We drew the design matrix $x$ from a normal distribution and held it fixed throughout. We set prior hyperparameters $\sigma_\beta^2 = 10$, $\alpha_\tau = 1$, and $\beta_\tau = 1$. To get the "ground truth" covariance matrix, we took 20000 draws from the posterior with the R MCMCglmm package [18], which used a combination of Gibbs and Metropolis Hastings sampling. Our LRVB estimates used the autodifferentiation software JuMP [19].

Results are shown in Fig. (1). Since $\tau$ is high in many of the simulations, $z$ and $\beta$ are correlated, and MFVB underestimates the standard deviation of $\beta$ and $\tau$. LRVB matches the MCMC standard deviation for all $\beta$, and matches for $\tau$ in all but the most correlated simulations. When $\tau$ gets very high, the MFVB assumption starts to bias the point estimates of $\tau$, and the LRVB standard deviations start to differ from MCMC. Even in that case, however, the LRVB standard deviations are much more accurate than the MFVB estimates, which underestimate the uncertainty dramatically. The final plot shows that LRVB estimates the covariances of $z$ with $\beta$, $\tau$, and $\log\tau$ reasonably well, while MFVB considers them independent.

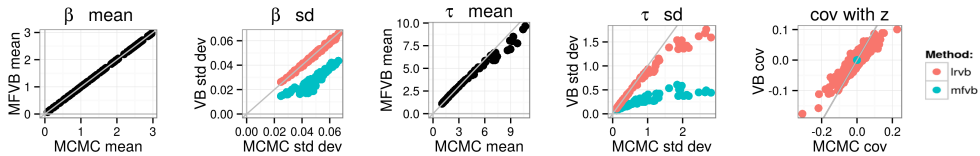

Figure 1: Posterior mean and covariance estimates on normal-Poisson simulation data.

## 3.2 Linear random effects

**Model.** Next, we consider a simple random slope linear model, with full details in Appendix E. We observe scalars $y_n$ and $r_n$ and a vector $x_n$, for $n = 1, ..., N$. Implicitly below, we will everywhere

condition on all the $x_n$ and $r_n$, which we consider to be fixed design matrices. In general, each random effect may appear in multiple observations, and the index $k(n)$ indicates which random effect, $z_k$, affects which observation, $y_n$. The full generative model is:

$$y_n|\beta, z, \tau \stackrel{indep}{\sim} \mathcal{N}\left(y_n|\beta^T x_n + r_n z_{k(n)}, \tau^{-1}\right), \quad z_k|\nu \stackrel{iid}{\sim} \mathcal{N}\left(z_k|0, \nu^{-1}\right),$$
$$\beta \sim \mathcal{N}(\beta|0, \Sigma_\beta), \quad \nu \sim \Gamma(\nu|\alpha_\nu, \beta_\nu), \quad \tau \sim \Gamma(\tau|\alpha_\tau, \beta_\tau).$$

We assume the mean-field factorization $q\left(\beta, \nu, \tau, z\right) = q\left(\beta\right) q\left(\tau\right) q\left(\nu\right) \prod_{k=1}^{K} q\left(z_n\right)$. Since this is a conjugate model, the optimal $q$ will be in the exponential family with no additional assumptions.

**Results.** We simulated 100 datasets of 300 datapoints each and 30 distinct random effects. We set prior hyperparameters to $\alpha_\nu = 2$, $\beta_\nu = 2$, $\alpha_\tau = 2$, $\beta_\tau = 2$, and $\Sigma_\beta = 0.1^{-1}I$. Our $x_n$ was 2-dimensional. As in Section 3.1, we implemented the variational solution using the autodifferentiation software JuMP [19]. The MCMC fit was performed with using MCMCglmm [18].

Intuitively, when the random effect explanatory variables $r_n$ are highly correlated with the fixed effects $x_n$, then the posteriors for $z$ and $\beta$ will also be correlated, leading to a violation of the mean field assumption and an underestimated MFVB covariance. In our simulation, we used $r_n = x_{1n} + \mathcal{N}(0, 0.4)$, so that $r_n$ is correlated with $x_{1n}$ but not $x_{2n}$. The result, as seen in Fig. (2), is that $\beta_1$ is underestimated by MFVB, but $\beta_2$ is not. The $\nu$ parameter, in contrast, is not well-estimated by the MFVB approximation in many of the simulations. Since the LRVB depends on the approximation $m_t^* \approx \mathbb{E}_{p_t}\theta$, its LRVB covariance is not accurate either (Fig. (2)). However, LRVB still improves on the MFVB standard deviation.

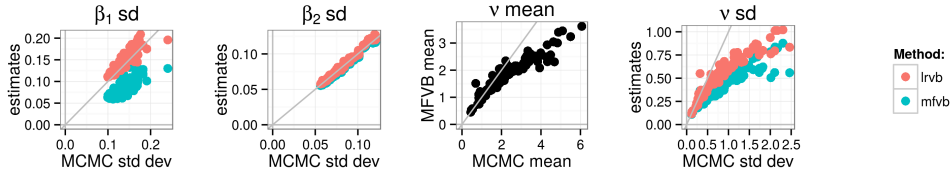

Figure 2: Posterior mean and covariance estimates on linear random effects simulation data.

### 3.3 Mixture of normals

**Model.** Mixture models constitute some of the most popular models for MFVB application [1, 2] and are often used as an example of where MFVB covariance estimates may go awry [5, 6]. Thus, we will consider in detail a Gaussian mixture model (GMM) consisting of a $K$-component mixture of $P$-dimensional multivariate normals with unknown component means, covariances, and weights. In what follows, the weight $\pi_k$ is the probability of the $k$th component, $\mu_k$ is the $P$-dimensional mean of the $k$th component, and $\Lambda_k$ is the $P \times P$ precision matrix of the $k$th component (so $\Lambda_k^{-1}$ is the covariance parameter). $N$ is the number of data points, and $x_n$ is the $n$th observed $P$-dimensional data point. We employ the standard trick of augmenting the data generating process with the latent indicator variables $z_{nk}$, for $n = 1, ..., N$ and $k = 1, ..., K$, such that $z_{nk} = 1$ implies $x_n \sim \mathcal{N}(\mu_k, \Lambda_k^{-1})$. So the generative model is:

$$P(z_{nk} = 1) = \pi_k, \quad p(x|\pi, \mu, \Lambda, z) = \prod_{n=1:N} \prod_{k=1:K} \mathcal{N}(x_n|\mu_k, \Lambda_k^{-1})^{z_{nk}} \tag{10}$$

We used diffuse conditionally conjugate priors (see Appendix F for details). We make the variational assumption $q\left(\mu, \pi, \Lambda, z\right) = \prod_{k=1}^{K} q\left(\mu_k\right) q\left(\Lambda_k\right) q\left(\pi_k\right) \prod_{n=1}^{N} q\left(z_n\right)$. We compare the accuracy and

speed of our estimates to Gibbs sampling on the augmented model (Eq. (10)) using the function rnmixGibbs from the R package bayesm. We implemented LRVB in C++, making extensive use of RcppEigen [20]. We evaluate our results both on simulated data and on the MNIST data set [21].

**Results.** For simulations, we generated $N = 10000$ data points from $K = 2$ multivariate normal components in $P = 2$ dimensions. MFVB is expected to underestimate the marginal variance of $\mu$, $\Lambda$, and $\log(\pi)$ when the components overlap since that induces correlation in the posteriors due to the uncertain classification of points between the clusters. We check the covariances estimated with Eq. (7) against a Gibbs sampler, which we treat as the ground truth.[3]

We performed $198$ simulations, each of which had at least $500$ effective Gibbs samples in each variable—calculated with the R tool effectiveSize from the coda package [22]. The first three plots show the diagonal standard deviations, and the third plot shows the off-diagonal covariances. Note that the off-diagonal covariance plot excludes the MFVB estimates since most of the values are zero. Fig. (3) shows that the raw MFVB covariance estimates are often quite different from the Gibbs sampler results, while the LRVB estimates match the Gibbs sampler closely.

For a real-world example, we fit a $K = 2$ GMM to the $N = 12665$ instances of handwritten 0s and 1s in the MNIST data set. We used PCA to reduce the pixel intensities to $P = 25$ dimensions. Full details are provided in Appendix G. In this MNIST analysis, the $\Lambda$ standard deviations were under-estimated by MFVB but correctly estimated by LRVB (Fig. (3)); the other parameter standard deviations were estimated correctly by both and are not shown.

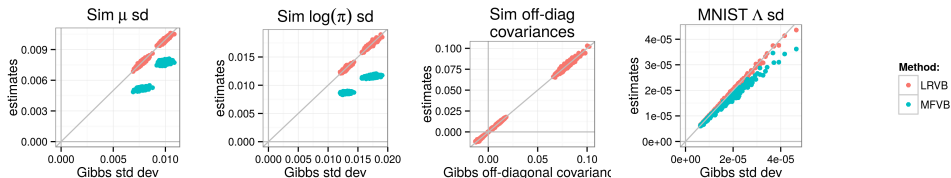

Figure 3: Posterior mean and covariance estimates on GMM simulation and MNIST data.

## 3.4 Scaling experiments

We here explore the computational scaling of LRVB in more depth for the finite Gaussian mixture model (Section 3.3). In the terms of Section 2.3, $\alpha$ includes the sufficient statistics from $\mu$, $\pi$, and $\Lambda$, and grows as $O(KP^2)$. The sufficient statistics for the variational posterior of $\mu$ contain the $P$-length vectors $\mu_k$, for each $k$, and the $(P+1)P/2$ second-order products in the covariance matrix $\mu_k \mu_k^T$. Similarly, for each $k$, the variational posterior of $\Lambda$ involves the $(P+1)P/2$ sufficient statistics in the symmetric matrix $\Lambda_k$ as well as the term $\log |\Lambda_k|$. The sufficient statistics for the posterior of $\pi_k$ are the $K$ terms $\log \pi_k$.[4] So, minimally, Eq. (7) will require the inverse of a matrix of size

$O(KP^2)$. The sufficient statistics for $z$ have dimension $K \times N$. Though the number of parameters thus grows with the number of data points, $H_z = 0$ for the multivariate normal (see Appendix F), so we can apply Eq. (8) to replace the inverse of an $O(KN)$-sized matrix with multiplication by the same matrix. Since a matrix inverse is cubic in the size of the matrix, the worst-case scaling for LRVB is then $O(K^2)$ in $K$, $O(P^6)$ in $P$, and $O(N)$ in $N$.

In our simulations (Fig. (4)) we can see that, in practice, LRVB scales linearly[5] in $N$ and approximately cubically in $P$ across the dimensions considered.[6] The $P$ scaling is presumably better than the theoretical worst case of $O(P^6)$ due to extra efficiency in the numerical linear algebra. Note that the vertical axis of the leftmost plot is on the log scale. At all the values of $N$, $K$ and $P$ considered here, LRVB was at least as fast as Gibbs sampling and often orders of magnitude faster.

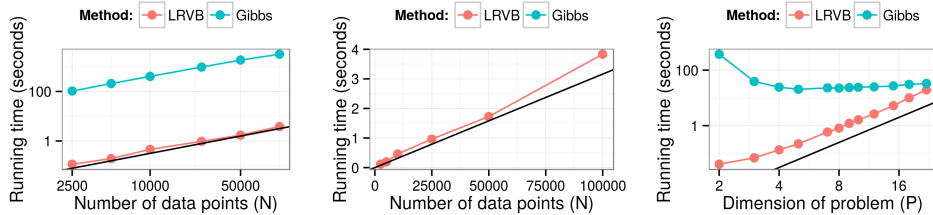

Figure 4: Scaling of LRVB and Gibbs on simulation data in both log and linear scales. Before taking logs, the line in the two lefthand (N) graphs is $y \propto x$, and in the righthand (P) graph, it is $y \propto x^3$.

## 4 Conclusion

The lack of accurate covariance estimates from the widely used mean-field variational Bayes (MFVB) methodology has been a longstanding shortcoming of MFVB. We have demonstrated that in sparse models, our method, linear response variational Bayes (LRVB), can correct MFVB to deliver these covariance estimates in time that scales linearly with the number of data points. Furthermore, we provide an easy-to-use formula for applying LRVB to a wide range of inference problems. Our experiments on a diverse set of models have demonstrated the efficacy of LRVB, and our detailed study of scaling of mixtures of multivariate Gaussians shows that LRVB can be considerably faster than traditional MCMC methods. We hope that in future work our results can be extended to more complex models, including Bayesian nonparametric models, where MFVB has proven its practical success.

**Acknowledgments.** The authors thank Alex Blocker for helpful comments. R. Giordano and T. Broderick were funded by Berkeley Fellowships.

## Footnotes

[1]For a comparison of this formula with the frequentist "supplemented expectation-maximization" procedure see Appendix C.

[2]All the code is available on our Github repository, rgiordan/LinearResponseVariationalBayesNIPS2015,

[3]The likelihood described in Section 3.3 is symmetric under relabeling. When the component locations and shapes have a real-life interpretation, the researcher is generally interested in the uncertainty of $\mu$, $\Lambda$, and $\pi$ for a particular labeling, not the marginal uncertainty over all possible re-labelings. This poses a problem for standard MCMC methods, and we restrict our simulations to regimes where label switching did not occur in our Gibbs sampler. The MFVB solution conveniently avoids this problem since the mean field assumption prevents it from representing more than one mode of the joint posterior.

[4]Since $\sum_{k=1}^{K} \pi_k = 1$, using $K$ sufficient statistics involves one redundant parameter. However, this does not violate any of the necessary assumptions for Eq. (7), and it considerably simplifies the calculations. Note that though the perturbation argument of Section 2 requires the parameters of $p(\theta|x)$ to be in the interior of the feasible space, it does not require that the parameters of $p(x|\theta)$ be interior.

[5]The Gibbs sampling time was linearly rescaled to the amount of time necessary to achieve 1000 effective samples in the slowest-mixing component of any parameter. Interestingly, this rescaling leads to increasing efficiency in the Gibbs sampling at low $P$ due to improved mixing, though the benefits cease to accrue at moderate dimensions.

[6]For numeric stability we started the optimization procedures for MFVB at the true values, so the time to compute the optimum in our simulations was very fast and not representative of practice. On real data, the optimization time will depend on the quality of the starting point. Consequently, the times shown for LRVB are only the times to compute the LRVB estimate. The optimization times were on the same order.

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
