[Supplementary Material]

# Supplementary Material

You can find this paper, as well as all the code necessary to run the described experiments, in our Github repo, rgiordan/LinearResponseVariationalBayesNIPS2015.

## A   LRVB estimates of the covariance of functions

In Section 2.2, we derived an estimate of the covariance of the natural sufficient statistics, $\theta$, of our variational approximation, $q(\theta)$. In this section we derive a version of Eq. (7) for the covariance of functions of $\theta$.

We begin by estimating the covariance between $\theta$ and a function $\phi(\theta)$. Suppose we have an MFVB solution, $q(\theta)$, to Eq. (1). Define the expectation of $\phi(\theta)$ to be $\mathbb{E}_q[\phi(\theta)] := f(m)$. This expectation is function of $m$ alone since $m$ completely parameterizes $q$. As in Eq. (3), we can consider a perturbed log likelihood that also includes $f(m)$:

$$
\begin{aligned}
\log p_t(\theta|x) &= \log p + t_0^T m + t_f f(m) := \log p + t^T m_f \\
t &:= \begin{pmatrix} t_0 \\ t_f \end{pmatrix} \qquad m_f := \begin{pmatrix} m \\ f(m) \end{pmatrix}
\end{aligned}
$$

Using the same reasoning that led to Eq. (4), we will define

$$
\Sigma_{\theta\phi} = \mathrm{Cov}_p(\theta, \phi(\theta)) \approx \frac{dm_t^*}{dt_f} =: \hat{\Sigma}_{\theta\phi}
$$

We then have the following lemma:

**Lemma A.1.** *If $\mathbb{E}_q[\phi(\theta)] =: f(m)$ is a differentiable function of $m$ with gradient $\nabla f$, then*

$$
\hat{\Sigma}_{\theta\phi} = \hat{\Sigma}\nabla f
$$

*Proof.* The derivative of the perturbed ELBO, $E_t$, is given by:

$$
\begin{aligned}
E_t &:= E + t^T m_f \\
\frac{\partial E_t}{\partial m} &= \frac{\partial E}{\partial m} + \begin{pmatrix} I & \nabla f \end{pmatrix} \begin{pmatrix} t_0 \\ t_f \end{pmatrix}
\end{aligned}
$$

The fixed point Eq. (2) then gives:

$$
\begin{aligned}
M_t(m) &:= M(m) + \begin{pmatrix} I & \nabla f \end{pmatrix} \begin{pmatrix} t_0 \\ t_f \end{pmatrix} \\
\frac{dm_t^*}{dt^T} &= \left.\frac{\partial M_t}{\partial m^T}\right|_{m=m_t^*} \frac{dm_t^*}{dt^T} + \frac{\partial M_t}{\partial t^T} \\
&= \left(\left.\frac{\partial M}{\partial m^T}\right|_{m=m_t^*} + \frac{\partial}{\partial m^T}\begin{pmatrix} I & \nabla f \end{pmatrix}\begin{pmatrix} t_0 \\ t_f \end{pmatrix}\right)\frac{dm^*}{dt^T} + \begin{pmatrix} I & \nabla f \end{pmatrix}
\end{aligned}
$$

The term $\frac{\partial}{\partial m^T} \begin{pmatrix} I & \nabla f \end{pmatrix} \begin{pmatrix} t_0 \\ t_f \end{pmatrix}$ is awkward, but it disappears when we evaluate at $t = 0$, giving

$$
\begin{aligned}
\frac{dm_t^*}{dt^T} &= \left( \left. \frac{\partial M}{\partial m^T} \right|_{m=m_t^*} \right) \frac{dm^*}{dt^T} + \begin{pmatrix} I & \nabla f \end{pmatrix} \\
&= \left( \frac{\partial^2 E}{\partial m \partial m^T} + I \right) \frac{dm^*}{dt^T} + \begin{pmatrix} I & \nabla f \end{pmatrix} \Rightarrow \\
\frac{dm^*}{dt^T} &= -\left( \frac{\partial^2 E}{\partial m \partial m^T} \right)^{-1} \begin{pmatrix} I & \nabla f \end{pmatrix}
\end{aligned}
$$

Recalling that

$$
\frac{dm^*}{dt_0^T} := \hat{\Sigma}
$$

We can plug in to see that

$$
\hat{\Sigma}_{\theta\phi} = \frac{dm^*}{dt_f} = \hat{\Sigma} \nabla f \tag{11}
$$

□

Finally, suppose we are interested in estimating $\mathrm{Cov}_p(\gamma(\theta), \phi(\theta))$, where $g(m) := \mathbb{E}_q [\gamma(\theta)]$. Again using the same reasoning that led to Eq. (4), we will define

$$
\Sigma_{\gamma\phi} = \mathrm{Cov}_p(\gamma(\theta), \phi(\theta)) \approx \frac{d\mathbb{E}_q [\gamma(\theta)]}{dt_f} =: \hat{\Sigma}_{\gamma\phi}
$$

**Proposition A.2.** *If* $\mathbb{E}_q [\phi(\theta)] = f(m)$ *and* $\mathbb{E}_q [\gamma(\theta)] = g(m)$ *are differentiable functions of* $m$ *with gradients* $\nabla f$ *and* $\nabla g$ *respectively, then*

$$
\hat{\Sigma}_{\gamma\phi} = \nabla g^T \hat{\Sigma} \nabla f
$$

*Proof.* By Lemma A.1 an application of the chain rule,

$$
\hat{\Sigma}_{\gamma\phi} = \frac{d\mathbb{E}_q [\gamma(\theta)]}{dt_f} = \frac{dg(m)}{dt_f} = \frac{dg(m)}{dm^T} \frac{dm}{dt_f} = \nabla g^T \hat{\Sigma} \nabla f
$$

□

## B  Exactness of LRVB for multivariate normal means

For any target distribution $p(\theta|x)$, it is well-known that MFVB cannot be used to estimate the covariances between the components of $\theta$. In particular, if $q^*$ is the estimate of $p(\theta|x)$ returned by MFVB, $q^*$ will have a block-diagonal covariance matrix—no matter the form of the covariance of $p(\theta|x)$.

Consider approximating a multivariate Gaussian posterior distribution $p(\theta|x)$ with MFVB. The Gaussian is the unique distribution that is fully determined by its mean and covariance. This posterior arises, for instance, given a multivariate normal likelihood $p(x|\mu) = \prod_{n=1:N} \mathcal{N}(x_n|\mu, S)$ with fixed covariance $S$ and an improper uniform prior on the mean parameter $\mu$. We make the mean field factorization assumption $q(\mu) = \prod_{d=1:D} q(\mu_d)$, where $D$ is the total dimension of $\mu$. This fact is often used to illustrate the shortcomings of MFVB [5–7]. In this case, it is well known that the MFVB posterior means are correct, but the marginal variances are underestimated if $S$ is not

diagonal. However, since the posterior means are correctly estimated, the LRVB approximation in Eq. (7) is in fact an equality. That is, for this model, $\hat{\Sigma} = dm_t/dt^T = \Sigma$ exactly.

In order to prove this result, we will rely on the following lemma.

**Lemma B.1.** *Consider a target posterior distribution characterized by $p(\theta|x) = \mathcal{N}(\theta|\mu, \Sigma)$, where $\mu$ and $\Sigma$ may depend on $x$, and $\Sigma$ is invertible. Let $\theta = (\theta_1, \ldots, \theta_J)$, and consider a MFVB approximation to $p(\theta|x)$ that factorizes as $q(\theta) = \prod_j q(\theta_j)$. Then the variational posterior means are the true posterior means; i.e. $m_j = \mu_j$ for all $j$ between $1$ and $J$.*

*Proof.* The derivation of MFVB for the multivariate normal can be found in Section 10.1.2 of [5]; we highlight some key results here. Let $\Lambda = \Sigma^{-1}$. Let the $j$ index on a row or column correspond to $\theta_j$, and let the $-j$ index correspond to $\{\theta_i : i \in [J] \setminus j\}$. E.g., for $j = 1$,

$$\Lambda = \left[ \begin{array}{cc} \Lambda_{11} & \Lambda_{1,-1} \\ \Lambda_{-1,1} & \Lambda_{-1,-1} \end{array} \right].$$

By the assumption that $p(\theta|x) = \mathcal{N}(\theta|\mu, \Sigma)$, we have

$$
\begin{aligned}
&\log p(\theta_j | \theta_{i \in [J] \setminus j}, x) \\
&= -\frac{1}{2}(\theta_j - \mu_j)^T \Lambda_{jj}(\theta_j - \mu_j) + (\theta_j - \mu_j)^T \Lambda_{j,-j}(\theta_{-j} - \mu_{-j}) + C,
\end{aligned}
\tag{12}
$$

where the final term is constant with respect to $\theta_j$. It follows that

$$
\begin{aligned}
\log q_j^*(\theta_j) &= \mathbb{E}_{q_i^*: i \in [J] \setminus j} \log p(\theta, x) + C \\
&= -\frac{1}{2}\theta_j^T \Lambda_{jj}\theta_j + \theta_j \mu_j \Lambda_{jj} - \theta_j \Lambda_{j,-j}(\mathbb{E}_{q^*}\theta_{-j} - \mu_{-j}).
\end{aligned}
$$

So

$$q_j^*(\theta_j) = \mathcal{N}(\theta_j | m_j, \Lambda_{jj}^{-1}),$$

with mean parameters

$$m_j = \mathbb{E}_{q_j^*}\theta_j = \mu_j - \Lambda_{jj}^{-1}\Lambda_{j,-j}(m_{-j} - \mu_{-j}) \tag{13}$$

as well as an equation for $\mathbb{E}_{q^*}\theta^T\theta$.

Note that $\Lambda_{jj}$ must be invertible, for if it were not, $\Sigma$ would not be invertible.

The solution $m = \mu$ is a unique stable point for Eq. (13), since the fixed point equations for each $j$ can be stacked and rearranged to give

$$
m - \mu = - \begin{bmatrix} 0 & \Lambda_{11}^{-1}\Lambda_{12} & \cdots & \Lambda_{11}^{-1}\Lambda_{1(J-1)} & \Lambda_{11}^{-1}\Lambda_{1J} \\ \vdots & & \ddots & & \vdots \\ \Lambda_{JJ}^{-1}\Lambda_{J1} & \Lambda_{JJ}^{-1}\Lambda_{J2} & \cdots & \Lambda_{JJ}^{-1}\Lambda_{J(J-1)} & 0 \end{bmatrix} (m - \mu)
$$

$$
= - \begin{bmatrix} \Lambda_{11}^{-1} & \cdots & 0 & \cdots & 0 \\ \vdots & \ddots & & & \vdots \\ 0 & & \ddots & & 0 \\ \vdots & & & \ddots & \vdots \\ 0 & \cdots & 0 & \cdots & \Lambda_{JJ}^{-1} \end{bmatrix} \begin{bmatrix} 0 & \Lambda_{12} & \cdots & \Lambda_{1(J-1)} & \Lambda_{1J} \\ \vdots & & \ddots & & \vdots \\ \Lambda_{J1} & \Lambda_{J2} & \cdots & \Lambda_{J(J-1)} & 0 \end{bmatrix} (m - \mu) \Leftrightarrow
$$

$$
0 = \begin{bmatrix} \Lambda_{11} & \cdots & 0 & \cdots & 0 \\ \vdots & \ddots & & & \vdots \\ 0 & & \ddots & & 0 \\ \vdots & & & \ddots & \vdots \\ 0 & \cdots & 0 & \cdots & \Lambda_{JJ} \end{bmatrix} (m - \mu) +
$$

$$
\begin{bmatrix} 0 & \Lambda_{12} & \cdots & \Lambda_{1(J-1)} & \Lambda_{1J} \\ \vdots & & \ddots & & \vdots \\ \Lambda_{J1} & \Lambda_{J2} & \cdots & \Lambda_{J(J-1)} & 0 \end{bmatrix} (m - \mu) \Leftrightarrow
$$

$$
0 = \Lambda (m - \mu) \Leftrightarrow
$$

$$
m = \mu.
$$

The last step follows from the assumption that $\Sigma$ (and hence $\Lambda$) is invertible. It follows that $\mu$ is the unique stable point of Eq. (13).

$\square$

**Proposition B.2.** *Assume we are in the setting of Lemma B.1, where additionally $\mu$ and $\Sigma$ are on the interior of the feasible parameter space. Then the LRVB covariance estimate exactly captures the true covariance, $\hat{\Sigma} = \Sigma$.*

*Proof.* Consider the perturbation for LRVB defined in Eq. (3). By perturbing the log likelihood, we change both the true means $\mu_t$ and the variational solutions, $m_t$. The result is a valid density function since the original $\mu$ and $\Sigma$ are on the interior of the parameter space. By Lemma B.1, the MFVB solutions are exactly the true means, so $m_{t,j} = \mu_{t,j}$, and the derivatives are the same as well. This means that the first term in Eq. (7) is not approximate, i.e.

$$
\frac{dm_t}{dt^T} = \frac{d}{dt^T} \mathbb{E}_{p_t} \theta = \Sigma_t,
$$

It follows from the arguments above that the LRVB covariance matrix is exact, and $\hat{\Sigma} = \Sigma$.

$\square$

## C  Comparison with supplemented expectation-maximization

The result in Appendix B about the multivariate normal distribution draws a connection between LRVB corrections and the "supplemented expectation-maximization" (SEM) method of [23]. SEM

is an asymptotically exact covariance correction for the EM algorithm that transforms the full-data Fisher information matrix into the observed-data Fisher information matrix using a correction that is formally similar to Eq. (7). In this section, we argue that this similarity is not a coincidence; in fact the SEM correction is an asymptotic version of LRVB with two variational blocks, one for the missing data and one for the unknown parameters.

Although LRVB as described here requires a prior (unlike SEM, which supplements the MLE), the two covariance corrections coincide when the full information likelihood is approximately log quadratic and proportional to the posterior, $p(\theta|x)$. This might be expected to occur when we have a large number of independent data points informing each parameter—i.e., when a central limit theorem applies and the priors do not affect the posterior. In the full information likelihood, some terms may be viewed as missing data, whereas in the Bayesian model the same terms may be viewed as latent parameters, but this does not prevent us from formally comparing the two methods.

We can draw a term-by-term analogy with the equations in [23]. We denote variables from the SEM paper with a superscript "$SEM$" to avoid confusion. MFVB does not differentiate between missing data and parameters to be estimated, so our $\theta$ corresponds to $(\theta^{SEM}, Y_{mis}^{SEM})$ in [23]. SEM is an asymptotic theory, so we may assume that $(\theta^{SEM}, Y_{mis}^{SEM})$ have a multivariate normal distribution, and that we are interested in the mean and covariance of $\theta^{SEM}$.

In the E-step of [23], we replace $Y_{mis}^{SEM}$ with its conditional expectation given the data and other $\theta^{SEM}$. This corresponds precisely to Eq. (13), taking $\theta_j = Y_{mis}^{SEM}$. In the M-step, we find the maximum of the log likelihood with respect to $\theta^{SEM}$, keeping $Y_{mis}^{SEM}$ fixed at its expectation. Since the mode of a multivariate normal distribution is also its mean, this, too, corresponds to Eq. (13), now taking $\theta_j = \theta^{SEM}$.

It follows that the MFVB and EM fixed point equations are the same; i.e., our $M$ is the same as their $M^{SEM}$, and our $\partial M/\partial m$ of Eq. (5) corresponds to the transpose of their $DM^{SEM}$, defined in Eq. (2.2.1) of [23]. Since the "complete information" corresponds to the variance of $\theta^{SEM}$ with fixed values for $Y_{OBS}^{SEM}$, this is the same as our $\Sigma_{q^*,11}$, the variational covariance, whose inverse is $I_{oc}^{-1}$. Taken all together, this means that equation (2.4.6) of [23] can be re-written as our Eq. (7).

$$V^{SEM} = I_{oc}^{-1} \left( I - DM^{SEM} \right)^{-1} \Rightarrow$$

$$\Sigma = V \left( I - \left( \frac{\partial M}{\partial m^T} \right)^T \right)^{-1} = \left( I - \frac{\partial M}{\partial m^T} \right)^{-1} V$$

## D  Normal-Poisson details

In this section, we use this model to provide a detailed, step-by-step description of a simple LRVB analysis.

The full joint distribution for the model in Eq. (9) is

$$\log p\left(y, z, \beta, \tau\right) = \sum_{n=1}^{N} \left( -\frac{1}{2}\tau z_n^2 + x_n \tau \beta z_n - \frac{1}{2}x_n^2 \tau \beta^2 - \frac{1}{2}\log \tau \right)$$

$$+ \sum_{n=1}^{N} \left( -\exp\left(z_n\right) + z_n y_n \right) - \frac{1}{2\sigma_\beta^2}\beta^2 + \left(\alpha_\tau - 1\right)\log \tau - \beta_\tau \tau + C$$

We find a mean-field approximation under the factorization $q\left(\beta, \tau, z\right) = q\left(\beta\right) q\left(\tau\right) \prod_{n=1}^{N} q\left(z_n\right)$. By inspection, the log joint is quadratic in $\beta$, so the optimal $q\left(\beta\right)$ will be Gaussian [5]. Similarly, the log joint is a function of $\tau$ only via $\tau$ and $\log \tau$, so the optimal $q\left(\tau\right)$ will be gamma. However,

the joint does not take a standard exponential family form in $z_n$:

$$\log p \left( z_n | y, \beta, \tau \right) = \left( x_n \tau \beta + y_n \right) z_n - \frac{1}{2} \tau z_n^2 - \exp \left( z_n \right) + C$$

The difficulty is with the term $\exp \left( z_n \right)$. So we make the further restriction that

$$q \left( z_n \right) = \mathcal{N} \left( \cdot \right) = q \left( z_n; \mathbb{E} \left[ z_n \right], \mathbb{E} \left[ z_n^2 \right] \right).$$

Fortunately, the troublesome term has an analytic expectation, as a function of the mean parameters, under this variational posterior:

$$\mathbb{E}_q \left[ \exp \left( z_n \right) \right] = \exp \left( \mathbb{E}_q \left[ z_n \right] + \frac{1}{2} \left( \mathbb{E}_q \left[ z_n^2 \right] - \mathbb{E}_q \left[ z_n \right]^2 \right) \right).$$

We can now write the variational distribution in terms of the following mean parameters:

$$m = \left( \mathbb{E}_q \left[ \beta \right], \mathbb{E}_q \left[ \beta^2 \right], \mathbb{E}_q \left[ \tau \right], \mathbb{E}_q \left[ \log \tau \right], \mathbb{E}_q \left[ z_1 \right], \mathbb{E}_q \left[ z_1^2 \right], ..., \mathbb{E}_q \left[ z_N \right], \mathbb{E}_q \left[ z_N^2 \right] \right)^T.$$

Calculating the LRVB covariance consists of roughly four steps:

1. finding the MFVB optimum $q^*$,
2. computing the covariance $V$ of $q^*$,
3. computing $H$, the Hessian of $L(m)$, for $q^*$, and
4. computing the matrix inverse and solving $\left( I - VH \right)^{-1} V$.

For step (1), the LRVB correction is agnostic as to how the optimum is found. In our experiments below, we follow a standard coordinate ascent procedure for MFVB [5]. We analytically update $q \left( \beta \right)$ and $q \left( \tau \right)$. Given $q \left( \beta \right)$ and $q \left( \tau \right)$, finding the optimal $q \left( z \right)$ becomes $N$ separate two-dimensional optimization problems; there is one dimension for each of the mean parameters $\mathbb{E}_q \left[ z_n \right]$ and $\mathbb{E}_q \left[ z_n^2 \right]$. In our examples, we solved these problems sequentially using IPOPT [24].

To compute $V$ for step (2), we note that by the mean-field assumption, $\beta$, $\tau$, and $z_n$ are independent, so $V$ is block diagonal. Since we have chosen convenient variational distributions, the mean parameters have known covariance matrices. For example, from standard properties of the normal distribution, $\text{Cov} \left( \beta, \beta^2 \right) = 2 \mathbb{E}_q \left[ \beta \right] \left( \mathbb{E}_q \left[ \beta^2 \right] - \mathbb{E}_q \left[ \beta \right]^2 \right)$.

For step (3), the mean parameters for $\beta$ and $\tau$ co-occur with each other and with all the $z_n$, so these four rows of $H$ are expected to be dense. However, the mean parameters for $z_n$ never occur with each other, so the bulk of $H$—the $2N \times 2N$ block corresponding to the mean parameters of $z$—will be block diagonal (Fig. (5b)). The Hessian of $L \left( m \right)$ can be calculated analytically, but we used the autodifferentiation software JuMP [19].

Finally, for step (4), we use the technique in Section 2.3 to exploit the sparsity of $V$ and $H$ (Fig. (5c)) in calculating $\left( I - VH \right)^{-1}$.

(a) MFVB covariance $V$        (b) Hessian matrix $H$        (c) $\left( I - VH \right)$

Figure 5: Sparsity patterns for $\hat{\Sigma} = \left( I - VH \right)^{-1}$ using the model in Eq. (9), $n = 5$ (white = 0)

# E    Random effects model details

As introduced in Section 3.2, our model is:

$$
\begin{aligned}
y_n | \beta, z, \tau & \overset{indep}{\sim} \mathcal{N}\left(\beta^T x_n + r_n z_{k(n)}, \tau^{-1}\right) \\
z_k | \nu & \overset{iid}{\sim} \mathcal{N}\left(0, \nu^{-1}\right)
\end{aligned}
$$

With the priors:

$$
\begin{aligned}
\beta & \sim \mathcal{N}\left(0, \Sigma_\beta\right) \\
\nu & \sim \Gamma\left(\alpha_\nu, \beta_\nu\right) \\
\tau & \sim \Gamma\left(\alpha_\tau, \beta_\tau\right)
\end{aligned}
$$

We will make the following mean field assumption:

$$
q\left(\beta, z, \tau, \nu\right) = q\left(\nu\right) q\left(\tau\right) q\left(\beta\right) \prod_{k=1}^{K} q\left(z_k\right)
$$

We have $n \in \{1, ..., N\}$, and $k \in \{1, ..., K\}$, and $k\left(n\right)$ matches an observation $n$ to a random effect $k$, allowing repeated observations of a random effect. The full joint log likelihood is:

$$
\begin{aligned}
\log p\left(y_n | z_{k(n)}, \tau, \beta\right) & = -\frac{\tau}{2}\left(y_n - \beta^T x_n - r_n z_{k(n)}\right)^2 + \frac{1}{2}\log \tau + C \\
\log p\left(z_k | \nu\right) & = -\frac{\nu}{2}z_k^2 + \frac{1}{2}\log \nu + C \\
\log p\left(\beta\right) & \quad -\frac{1}{2}\text{trace}\left(\Sigma_\beta^{-1}\beta\beta^T\right) + C \\
\log p\left(\tau\right) & = \left(\alpha_\tau - 1\right)\log \tau - \beta_\tau \tau + C \\
\log p\left(\nu\right) & = \left(\alpha_\nu - 1\right)\log \nu - \beta_\nu \nu + C \\
\log p\left(y, \tau, \beta, z\right) & = \sum_{n=1}^{N}\log p\left(y_n | z_{k(n)}, \tau, \beta\right) + \sum_{k=1}^{K}\log p\left(z_k | \nu\right) + \\
& \quad \log p\left(\beta\right) + \log p\left(\nu\right) + \log p\left(\tau\right)
\end{aligned}
$$

Expanding the first term of the conditional likelihood of $y_n$ gives

$$
\begin{aligned}
& -\frac{\tau}{2}\left(y_n - \beta^T x_n - r_n z_{k(n)}\right)^2 \\
= & -\frac{\tau}{2}\left(y_n^2 - 2y_n x_n^T \beta - 2y_n r_n z_{n(k)} + \text{trace}\left(x_n x_n^T \beta\beta^T\right) + r_n^2 z_{k(n)}^2 + 2r_n x_n^T \beta z_{k(n)}\right)
\end{aligned}
$$

By grouping terms, we can see that the mean parameters will be

$$
\begin{aligned}
q\left(\beta\right) & = q\left(\beta; \mathbb{E}_q\left[\beta\right], \mathbb{E}_q\left[\beta\beta^T\right]\right) \\
q\left(z_k\right) & = q\left(z_k; \mathbb{E}_q\left[z_k\right], \mathbb{E}_q\left[z_k^2\right]\right) \\
q\left(\tau\right) & = q\left(\tau; \mathbb{E}_q\left[\tau\right], \mathbb{E}_q\left[\log \tau\right]\right) \\
q\left(\nu\right) & = q\left(\nu; \mathbb{E}_q\left[\nu\right], \mathbb{E}_q\left[\log \nu\right]\right)
\end{aligned}
$$

It follows that the optimal variational distributions are $q\left(\beta\right)$ =multivariate normal, $q\left(z_k\right)$ =univariate normal, and $q\left(\tau\right)$ and $q\left(\nu\right)$ will be gamma. We performed standard coordinate ascent on these distributions [5].

As in Section 3.1, we implemented this model in the autodifferentiation software JuMP [19]. This means conjugate coordinate updates were easy, since the natural parameters corresponding to a mean

parameters are the first derivatives of the log likelihood with respect to the mean parameters. For example, denoting the log likelihood at step $s$ by $L_s$, the update for $q_{s+1}(z_k)$ will be:

$$\log q_{s+1}(z_k) = \frac{\partial \mathbb{E}_q[L_s]}{\partial \mathbb{E}_q[z_k]} z_k + \frac{\partial \mathbb{E}_q[L_s]}{\partial \mathbb{E}_q[z_k^2]} z_k^2 + C$$

Given the partial derivatives of $L_s$ with respect to the mean parameters, the updated mean parameters for $z_k$ can be read off directly using standard properties of the normal distribution.

The variational covariance matrices are all standard. We can see that $H$ will have nonzero terms in general (for example, the three-way interaction $\mathbb{E}_q[\tau]\,\mathbb{E}_q[z_{k(n)}]\,\mathbb{E}_q[\beta]$), and that LRVB will be different from MFVB. As usual in our models, $H$ is sparse, and we can easily apply the technique in section Section 2.3 to get the covariance matrix excluding the random effects, $z$.

## F  Multivariate normal mixture details

In this section we derive the basic formulas needed to calculate Eq. (7) for a finite mixture of normals, which is the model used in Section 3. We will follow the notation introduced in Section 3.3.

Let each observation, $x_n$, be a $P \times 1$ vector. We will denote the $P$th component of the $n$th observation $x_n$, with a similar pattern for $z$ and $\mu$. We will denote the $p, q$th entry in the matrix $\Lambda_k$ as $\Lambda_{k,pq}$. The data generating process is as follows:

$$P(x|\mu, \pi, \Lambda) = \prod_{n=1}^{N} P(x_n|z_n, \mu, \Lambda) \prod_{k=1}^{K} P(z_{nk}|\pi_k)$$

$$\log P(x_n|z_n, \mu, \Lambda) = \sum_{n=1}^{N} z_{nk} \log \phi_k(x_n) + C$$

$$\log \phi_k(x) = -\frac{1}{2}(x - \mu_k)^T \Lambda_k (x - \mu_k) + \frac{1}{2}\log|\Lambda_k| + C$$

$$\log P(z_{nk}|\pi_k) = \sum_{k=1}^{K} z_{nk} \log \pi_k + C$$

It follows that the log posterior is given by

$$\log P(z, \mu, \pi, \Lambda|x) = \sum_{n=1}^{N}\sum_{k=1}^{K} z_{nk}\left(\log \pi_k - \frac{1}{2}(x_n - \mu_k)^T \Lambda_k (x_n - \mu_k) + \frac{1}{2}\log|\Lambda_k|\right) +$$

$$\sum_{k=1}^{K}\log p(\mu_k) + \sum_{k=1}^{K}\log p(\Lambda_k) + \log p(\pi) + C$$

We used a multivariate normal prior for $\mu_k$, a Wishart prior for $\Lambda_k$, and a Dirichlet prior for $\pi$. In the simulations described in Section 3.3, we used the following prior parameters for the VB model:

$$p(\mu_k) = \mathcal{N}\left(0_P, \text{diag}_P(0.01)^{-1}\right)$$
$$p(\Lambda_k) = \text{Wishart}(\text{diag}_P(0.01), 1)$$
$$p(\pi) = \text{Dirichlet}(5_K)$$

Here, $\text{diag}_P(a)$ is a $P$-dimensional diagonal matrix with $a$ on the diagonal, and $0_P$ is a length $P$ vector of the value 0, with a similar definition for $5_K$. Unfortunately, the function we used for the MCMC calculations, rnmixGibbs in the package bayesm, uses a different form for the $\mu_k$ prior. Specifically, rnmixGibbs uses the prior

$$p_{MCMC}(\mu_k|\Lambda_k) = \mathcal{N}(0, a^{-1}\Lambda_k^{-1})$$

where $a$ is a scalar. There is no way to exactly match $p_{MCMC}(\mu_k)$ to $p(\mu_k)$, so we simply set $a = 0.01$. Since our datasets are all reasonably large, the prior was dominated by the likelihood, and we found the results extremely insensitive to the prior on $\mu_k$, so this discrepancy is of no practical importance.

The parameters $\mu_k$, $\Lambda_k$, $\pi$, and $z_n$ will each be given their own variational distribution. For $q_{\mu_k}$ we will use a multivariate normal distribution; for $q_{\Lambda_k}$ we will us a Wishart distirbution; for $q_\pi$ we will use a Dirichlet distribution; for $q_{z_n}$ we will use a Multinoulli (a single multinomial draw). These are all the optimal variational choices given the mean field assumption and the conditional conjugacy in the model.

The sufficient statistics for $\mu_k$ are all terms of the form $\mu_{kp}$ and $\mu_{kp}\mu_{kq}$. Consequently, the sub-vector of $\theta$ corresponding to $\mu_k$ is

$$\theta_{\mu_k} = \begin{pmatrix} \mu_{k1} \\ \vdots \\ \mu_{kp} \\ \mu_{k1}\mu_{k1} \\ \mu_{k1}\mu_{k2} \\ \vdots \\ \mu_{kP}\mu_{kP} \end{pmatrix}$$

We will only save one copy of $\mu_{kp}\mu_{kq}$ and $\mu_{kq}\mu_{kp}$, so $\theta_{\mu_k}$ has length $P + \frac{1}{2}(P+1)P$. For all the parameters, we denote the complete stacked vector without a $k$ subscript:

$$\theta_\mu = \begin{pmatrix} \theta_{\mu_1} \\ \vdots \\ \theta_{\mu_K} \end{pmatrix}$$

The sufficient statistics for $\Lambda_k$ are all the terms $\Lambda_{k,pq}$ and the term $\log|\Lambda_k|$. Again, since $\Lambda$ is symmetric, we do not keep redundant terms, so $\theta_{\Lambda_k}$ has length $1 + \frac{1}{2}(P+1)P$. The sufficient statistic for $\pi$ is the $K$-vector $(\log\pi_1, ..., \log\pi_K)$. The sufficient statistics for $z$ are simply the $N \times K$ values $z_{nk}$ themselves.

In terms of Section 2.3, we have

$$\alpha = \begin{pmatrix} \theta_\mu \\ \theta_\Lambda \\ \theta_\pi \end{pmatrix}$$
$$z = (\ \theta_z\ )$$

That is, we are primarily interested in the covariance of the sufficient statistics of $\mu$, $\Lambda$, and $\pi$. The latent variables $z$ are nuisance parameters.

To put the log likelihood in terms useful for LRVB, we must express it in terms of the sufficient statistics, taking into account the fact the $\theta$ vector does not store redundant terms (e.g. it will only

keep $\Lambda_{ab}$ for $a < b$ since $\Lambda$ is symmetric).

$$-\frac{1}{2} (x_n - \mu_k)^T \Lambda_k (x_n - \mu_k)$$

$$= -\frac{1}{2} \text{trace} \left( \Lambda_k (x_n - \mu_k)(x_n - \mu_k)^T \right)$$

$$= -\frac{1}{2} \sum_a \sum_b \left( \Lambda_{k,ab} (x_{n,a} - \mu_{k,a})(x_{n,b} - \mu_{k,b}) \right)$$

$$= -\frac{1}{2} \sum_a \sum_b \left( \Lambda_{k,ab} \mu_{k,a} \mu_{k,b} - \Lambda_{k,ab} x_{n,a} \mu_{k,b} - \Lambda_{k,ab} x_{n,b} \mu_{k,a} + \Lambda_{k,ab} x_{n,a} x_{n,b} \right)$$

$$= -\frac{1}{2} \sum_a \Lambda_{k,aa} \left( \mu_k^2 \right)^a + \sum_a \Lambda_{k,aa} x_{n,a} \mu_{k,a} - \frac{1}{2} \sum_a \Lambda_{k,aa} \left( x_n^2 \right)^2 -$$

$$\frac{1}{2} \sum_{a \neq b} \Lambda_{k,ab} \mu_{k,a} \mu_{k,b} + \sum_{a \neq b} \Lambda_{k,ab} x_{n,a} \mu_{k,b} - \frac{1}{2} \sum_{a \neq b} \Lambda_{k,ab} x_{n,a} x_{n,b}$$

$$= -\frac{1}{2} \sum_a \Lambda_{k,aa} \left( \mu_k^2 \right)^a + \sum_a \Lambda_{k,aa} x_{n,a} \mu_{k,a} - \frac{1}{2} \sum_a \Lambda_{k,aa} \left( x_n^2 \right)^2 -$$

$$\sum_{a < b} \Lambda_{k,ab} \mu_{k,a} \mu_{k,b} + \sum_{a < b} \Lambda_{k,ab} \left( x_{n,a} \mu_{k,b} + x_{n,b} \mu_{k,a} \right) - \sum_{a < b} \Lambda_{k,ab} x_{n,a} x_{n,b}$$

The MFVB updates and covariances in $V$ are all given by properties of standard distributions. To compute the LRVB corrections, it only remains to calculate the Hessian, $H$. These terms can be read directly off the posterior. First we calculate derivatives with respect to components of $\mu$.

$$\frac{\partial^2 H}{\partial \mu_{k,a} \partial \Lambda_{k,ab}} = \sum_i z_{nk} x_{n,b}$$

$$\frac{\partial^2 H}{\partial \left( \mu_{k,a} \mu_{k,b} \right) \partial \Lambda_{k,ab}} = -\left( \frac{1}{2} \right)^{1(a=b)} \sum_n z_{nk}$$

$$\frac{\partial^2 H}{\partial \mu_{k,a} \partial z_{nk}} = \sum_b \Lambda_{k,ab} x_{n,b}$$

$$\frac{\partial^2 H}{\partial \left( \mu_{k,a} \mu_{k,b} \right) \partial z_{nk}} = -\left( \frac{1}{2} \right)^{1(a=b)} \Lambda_{k,ab}$$

All other $\mu$ derivatives are zero. For $\Lambda$,

$$\frac{\partial^2 H}{\partial \Lambda_{k,ab} \partial z_{nk}} = -\left( \frac{1}{2} \right)^{1(a=b)} \left( x_{n,a} x_{n,b} - \mu_{k,a} x_{n,b} - \mu_{k,b} x_{n,a} + \mu_{k,a} \mu_{k,b} \right)$$

$$\frac{\partial^2 H}{\partial \log |\Lambda_k| \partial z_{nk}} = \frac{1}{2}$$

The remaining $\Lambda$ derivatives are zero. The only nonzero second derivatives for $\log \pi$ are to $Z$ and are given by

$$\frac{\partial^2 H}{\partial \log \pi_j \partial z_{nk}} = 1$$

Note in particular that $H_{zz} = 0$, allowing efficient calculation of Eq. (8).

## G  MNIST details

For a real-world example, we applied LRVB to the unsupervised classification of two digits from the MNIST dataset of handwritten digits. We first preprocess the MNIST dataset by performing principle component analysis on the training data's centered pixel intensities and keeping the top 25 components. For evaluation, the test data is projected onto the same 25-dimensional subspace found using the training data.

We then treat the problem of separating handwritten 0s from 1s as an unsupervised clustering problem. We limit the dataset to instances labeled as 0 or 1, resulting in 12665 training and 2115 test points. We fit the training data as a mixture of multivariate Gaussians. Here, $K = 2$, $P = 25$, and $N = 12665$. Then, keeping the $\mu$, $\Lambda$, and $\pi$ parameters fixed, we calculate the expectations of the latent variables $z$ in Eq. (10) for the test set. We assign test set data point $x_n$ to whichever component has maximum a posteriori expectation. We count successful classifications as test set points that match their cluster's majority label and errors as test set points that are different from their cluster's majority label. By this measure, our test set error rate was $0.08$. We stress that we intend only to demonstrate the feasibility of LRVB on a large, real-world dataset rather than to propose practical methods for modeling MNIST.