[Reviews · NeurIPS 2015]

Submitted by Assigned_Reviewer_1

The paper addresses the problem of estimating the uncertainty of model variables in a meanfield variational Bayes setting. The authors propose an extension of meanfield VB called linear response VB. The idea is to perturbate the posterior such that resulting posterior mean is unaffected and the posterior covariance estimate improves.

The authors present how to exploit linear response methods for meanfield VB and discuss how this effects various models. In experiments, the authors demonstare the effectiveness of the proposed extension which often reaches the accuracy of Gipps sampling at much lower computational cost.

Overall, the paper is well-written and nice to read. The only concern I have regards the contribution. Linear response methods have been studied before in the context of VB. However, I believe that especially the empirical findings are worth publishing.

minor issues: line 150: \mu(m) rather then \mu Figures 1 and 2 are very small
Summary: The authors extend meanfield variational Bayes by linear response methods to better estimate the uncertainty of the model variables. Since this idea is not new, the contribution may be debatable.

Submitted by Assigned_Reviewer_2

I think the basic idea of the approximation presented by the authors is very interesting.

However it is hard to appreciate the broad applicability of the idea from the presentations in Sections 2.1 and 2.2.

The presentation of the idea in the text differs a bit from what needs to be done to apply the method in the examples.

There is a model with parameter theta, and then we're told that the variational distribution q(theta) is in the exponential family with log q(theta)=eta^T theta-A(eta).

Now, for most exponential families (such as the multivariate Gaussian family) the sufficient statistic would be say t(theta) where the dimension of t(theta) is larger than theta.

This is problematic for the explanation of the method presented, since the authors go on to parametrize the lower bound in terms of E(theta) rather than eta, which would require theta and eta to be of the same dimension.

I think what the authors mean here is that the lower bound can be parametrized in terms of m=E(theta) and some additional parameters lambda say, and then for any fixed m the additional parameters can be set to their optimum values for the given m according to the lower bound.

By this kind of "profiling" we obtain a function of m only and then I think the arguments they give go through for this new profiled lower bound.

Perhaps the authors think this is an obvious technicality but it confused me for a while and I think would confuse other readers too.

I think it's also true that if one is interested in marginal inferences on a subset of the parameters the same idea can be applied to get a function from the lower bound depending only on the mean parameters for the parameters of interest.

This is something like a variational Bayes version of profile likelihood I guess.

I would conjecture that this kind of profiling focusing only on the parameters of interest increases the accuracy of the method compared to the method presented which deals with all the parameters at once.

A minor error I noted was that in the equation above (6), it should be -(-V^{-1}+H)^{-1} not (V^{-1}+H)^{-1}.
Summary: I think this is a useful and broadly applicable idea for improved quantification of uncertainty in mean field variational Bayes approximations.

I find the presentation of the basic idea confusing and I think that can be improved.

Submitted by Assigned_Reviewer_3

This paper derives covariance estimates for mean field variational Bayes using linear response theory. Mean field variational Bayes generally underestimates the variances, and provides no estimate of the covariance between latent variables in different factors. The authors propose an estimator of the posterior covariances for exponential family models based on generalizing linear response methods. The authors then present experiments across several models comparing to MCMC methods as a baseline.

My main concern with this work is that the presentation suggests that this technique can be used to calculate covariances whenever the approximating family is in the exponential family, rather to me it seems a bit more limited in that the approximating family needs to have the identity sufficient statistic (T(\theta) includes \theta). All of the examples presented fall into this subclass, but models with Dirichlet or Beta approximations.

It would nice if Equation 3 drew directly tied to exponential family forms. Something like the introduction of t makes a one parameter exponential family with the posterior as a base measure.

Some minor comments: - "Let V denote the mean vector and the covariance matrix of  " -> Unclear

- "horizotal" -> spelling

- It would be nice if the dimensionality in equation 4 was made more explicit through notation

- It would good to have a discussion of some of the more recent developments in variational Bayes that go beyond mean-field
Summary: This paper presents an intriguing approach to posterior covariance calculations using linear response theory for exponential family approximations. To me the presentation and limitations are not presented well enough to merit acceptance.

Submitted by Assigned_Reviewer_4

Summary: Linear response theory stems from theoretical physics where it is typically used to calculate conductivities, i.e., linear reactions of a system to an applied perturbation, such as a voltage. Here, the authors propose to follow

the same logic in a statistical context: a linear perturbation is added to the log posterior, and then second derivatives of the log partition function are taken with respect to the coupling parameter to obtain the response kernel. The authors derive a self-consistency equation whose solution gives an estimate to the posterior covariance. It is provably exact for

multivariate Gaussian posteriors, and a good approximation in other cases, as the authors demonstrate experimentally on various models.

This work is based on earlier work by Opper and Winther, but is generalized to exponential family variational posteriors. It is written in a very accessible way.

Quality: This is a very paper, accept if possible. It contains the right amount of derivations and empirical studies.

Originality: The paper contains original work, although the idea of linear response is not completely new (as discussed above).

Significance: This paper could be potentially significant when covariance estimates are of interest. It would be interesting for the readers

to comment on scalable stochastic variants of the approach.

Clarity: The writing style is very clear.
Summary: The paper discusses linear response for exponential family models,

a systematic way to go beyond mean-field variational inference to better approximate posterior covariances. Good paper, accept.

Author Feedback
Author rebuttal: We would like to thank the reviewers for their thorough and thoughtful comments. We will fix the minor typos in the paper without further comment here and use this space to respond to the reviewers' broader concerns.

First, one reviewer expressed concerns about the originality of LRVB. It is certainly true that the notion of perturbing a MFVB solution to recover covariance matrices has a long pedigree. We see our contribution not as introducing the idea itself, but clarifying it, expanding its scope to models of modern interest, and demonstrating its applicability to a wide range of problems for which it had not been previously considered. Many details of successfully implementing an LRVB solution are not immediately obvious from reading the previous literature (indeed, some of the reviewers' concerns and questions point to the subtlety of LRVB in general practice). The most recent work on this powerful technique was last published 2004 [14] but surveys (e.g. [11]) continue to point out the inadequacy of MFVB covariances without reference to linear response theory. More recently, David Dunson made a case for using MCMC over VB in his 2014 ISBA talk by arguing that variational methods exhibit "extreme under-estimation of posterior variance and lack of characterization of posterior dependence" (http://bayesian.org/sites/default/files/Dunson.pdf). In "On Markov chain Monte Carlo methods for tall data" (2015), authors Bardenet, Doucet, and Holmes dismiss variational methods as not "fully Bayesian" since they lack a "precise quantification of uncertainties." We address exactly these concerns in the current work.

Several reviewers expressed interest in connecting LRVB to other contemporary trends in variational inference, which is an interest that the authors share and have been exploring since submission. To begin with, we stress that LRVB augments an existing MFVB solution, and should work no matter what method is used to reach that solution. Although we used batch methods in our paper, LRVB should apply immediately to stochastic variational inference. Note that the cubic scaling in parameters is particular to LRVB and would not be solved by using a different MFVB solver, though we are interested in pursuing randomized matrix algorithms to mitigate the computational cost of inverting the Hessian. Since the current paper was submitted, we have experimented with incorporating non-analytic expectations into LRVB using Monte Carlo estimates with the "reparameterization trick" (e.g. [Kingma and Welling, 2014]). We are convinced that this exposition of LRVB is only the starting point for a broad and exciting range of fast, scalable inferential procedures.

Finally, we would like to clarify some issues related to how LRVB depends on the choice of sufficient statistic, which certainly deserves further explication in the final draft. One is free to use LRVB with variational approximations involving non-minimal sufficient statistics which impose constraints among the moment parameters. These constraints must, of course, be respected when finding the MFVB solution, and will be part of the fixed point equation used for LRVB. Given an MFVB optimum, LRVB only requires that the resulting variational approximation be a well-defined distribution in an open neighborhood of the optimal natural parameters so that solutions to the perturbed ELBO are well-defined. Note that several of our applications use non-minimal sufficient statistics, and we will be happy to expand on these examples in the appendix to clarify this point.

Reviewer 5 raised the interesting question of whether LRVB works when some parameters are not necessarily moment parameters of a variational distribution (e.g. when they parameterize a lower bound on an intractable ELBO term as in [Jaakola and Jordan, 1996]). In fact, the derivation of the LRVB correction would remain essentially unchanged except for a minor modification in the entropy derivative. This suggests a potential measure of "uncertainty" measures on non-stochastic terms and is an interesting avenue for further investigation.

Reviewers 2 and 5 both raised concerns about whether LRVB is capable of estimating covariances of quantities other than sufficient statistics. The answer is "yes" -- to estimate the covariance of two smooth functions, f, and g, one can measure the change in E_q[f(\theta)] with respect to a perturbation t*g(\theta) in the ELBO. This perturbation ends up being a linear combination of the LRVB covariances of the natural sufficient statistics, with the linear combination determined by the first order terms of a Taylor expansion of E_q[f(\theta)] and E_q[g(\theta)] as a function of the moment parameters of q. Of course, as with the rest of LRVB, the accuracy of the approximation depends on the accuracy of the original solution. We agree that this extension would be a valuable inclusion, and will be happy to put a derivation and proof in an appendix.